# Surgical Strategies to Promote Cutaneous Healing

**DOI:** 10.3390/medsci9020045

**Published:** 2021-06-16

**Authors:** Ines Maria Niederstätter, Jennifer Lynn Schiefer, Paul Christian Fuchs

**Affiliations:** Clinic of Plastic, Reconstructive, Hand and Burn Surgery, Hospital Cologne Merheim, University of Witten/Herdecke, 51109 Cologne, Germany; schieferj@kliniken-koeln.de (J.L.S.); fuchsp@kliniken-koeln.de (P.C.F.)

**Keywords:** wound healing, debridement, plastic surgery, reconstructive ladder, split-thickness skin graft, buried skin graft, micrograft

## Abstract

Usually, cutaneous wound healing does not get impeded and processes uneventfully, reaching wound closure easily. The goal of this repair process is to restore the integrity of the body surface by creating a resilient and stable scar. Surgical practice and strategies have an impact on the course of wound healing and the later appearance of the scar. By considering elementary surgical principles, such as the appropriate suture material, suture technique, and timing, optimal conditions for wound healing can be created. Wounds can be differentiated into clean wounds, clean–contaminated wounds, contaminated, and infected/dirty wounds, based on the degree of colonization or infection. Furthermore, a distinction is made between acute and chronic wounds. The latter are wounds that persist for longer than 4–6 weeks. Care should be taken to avoid surgical site infections in the management of wounds by maintaining sterile working conditions, using antimicrobial working techniques, and implementing the principles of preoperative antibiotics. Successful wound closure is influenced by wound debridement. Wound debridement removes necrotic tissue, senescent and non-migratory cells, bacteria, and foreign bodies that impede wound healing. Additionally, the reconstructive ladder is a viable and partially overlapping treatment algorithm in plastic surgery to achieve successful wound closure.

## 1. Introduction

Wound healing, defined as the cycle of tissue damage repair, is a process that everyone goes through repeatedly throughout their lives. Usually, cutaneous wound healing does not get impeded and processes uneventfully, reaching wound closure easily [1,2]. It is a perfectly timed process consisting of a complex collaboration of multiple cell strains and their products. At the bio-physiological level, the healing process begins with the formation of the wound and proceeds in three phases that occur both sequentially and overlap: inflammation, granulation, and remodeling of the scar [3]. An interplay of growth factors leads to cell proliferation and the integration of dynamic changes involving soluble mediators, blood cells, extracellular matrix production and proliferation of parenchymal cells [4]. This review article aims to highlight the different surgical principles to wound care and attempts to provide an overview of the options for wound care in plastic surgery.

Different books dealing with wound treatment and defect reconstruction as well as the electronic database Pubmed were used for the literature research.

## 2. Basics of Surgical Wound Care

A wound is defined as a trauma-caused tissue interruption or external or internal substance defect of a tissue with loss of tissue cohesion. Wounds can be caused by mechanical, thermal, chemical, and radiogenic damage. These trauma-caused wounds must be differentiated from iatrogenic wounds, resulting from surgical procedures. Wounds of the skin and the mucous membranes resulting from underlying chronic diseases such as diabetes mellitus, chronic venous/arterial insufficiency and immunological or dermatological diseases must be differentiated. These are considered chronic wounds [5,6].

From a surgical point of view, wound healing processes when the margins of a clean, surgical wound get adapted by tension-free sutures, staples or glue [7]. It requires an uninfected wound with even wound margins. Under these conditions wound closure occurs unimpededly through continuous replenishment by connective tissue [4,8]. This process, defined as wound healing by primary intention, can be achieved through en-bloc excision, debridement of the wound margins and closing the wound by sutures directly [9].

If sutures are not used to close the wound, secondary wound healing may be considered. These are wounds characterized by complete destruction of the epidermis and dermis as well as deeper structures [4]. In this process, wound healing occurs as tissue loss is replaced by granulation tissue [10]. Granulation tissue transforms into scar tissue. After contraction of the scar tissue, epithelial covering occurs. The endurance of secondary wound healing may take several weeks and depends on the wound defect dimensions. The goal of this repair process is to restore the integrity of the body surface by creating a resilient and stable scar [4]. In these wounds, which are not supported by sutures, depending on the wound’s localization, a constant stretching of the scar can be observed. The scar may double in size in the period from 3 to 12 weeks after incision. Further widening of the scar by 50% is possible in the period of 3 to 6 months after incision. Mechanical relief of the scars during this period is critical [11].

The appearance of the final scar depends on multiple factors: skin type, localization on the body, suture material and technique, wound alignment, and tension of the wound margins. Surgical practice and strategies have an impact on the course of wound healing and the later appearance of the scar. From a plastic surgeon’s perspective, it is desirable to minimize the negative effects of surgical and mechanical interventions by using the following surgical strategies [12].

### 2.1. Wound Localization and Wound Alignment

Wounds that are located on moved anatomical regions, for example joints, or on special regions as the breastbone, have a tendency to hypertrophy and widening. This, in turn, leads to unfavorable scarring [12].

Surgical skin incisions should be conducted with the scalpel in a perpendicular direction to create a clean incision edge. By placing them parallel to the “relaxed skin tension lines”, also called “Langer lines”, the most satisfactory aesthetic result can be achieved. The relaxed skin tension lines (Figure 1) indicate the direction of orientation of collagen fibers in the reticular layer of the skin. They run parallel to the principal muscle fibers below the skin. By placing the surgical incisions parallel to the relaxed tension lines only minimal tension on the wound edges is established. In this way, widening of the wound and hypertrophy of the scar can be minimized [13]. In general, tension should be avoided to counteract infection, necrosis, excessive scarring, and wound dehiscence [11,14].

### 2.2. Timing

Timing of the wound closure is a crucial aspect to consider. When closing a wound with sutures, 6 to 8 h should not be exceeded. Otherwise, the risk of infection increases. Highly perfused areas such as the scalp and the face are excepted. They can be closed with sutures up to 24 h after the injury. The degree of contamination with bacteria and foreign matter is a decisive point influencing wound healing [9,15,16,17].

### 2.3. Suture Material

One characteristic of modern suture materials and suturing techniques is to trigger as little inflammatory reaction as possible during wound healing. In general, with regard to non-absorbable suture-material, staples produce more inflammatory reactions than non-absorbable sutures. In addition, wound closure by staples ends more likely in unfavorable scarring [12]. Comparison of absorbable and nonabsorbable sutures shows a similar risk of wound infection and other postoperative complications for both materials. Absorbable sutures will most likely result in less wound dehiscence [18]. There is no difference in scar quality between absorbable and non-absorbable sutures [12]. With the use of monofilamentous absorbable sutures, fewer tissue reactions occur during wound healing than with braided absorbable sutures. According to Niessen et al., monofilamentous absorbable sutures provoke less reactive scars with a smaller probability of hypertrophic scar formation than braided absorbable sutures [19].

### 2.4. Suture Technique

Principles of wound closure include proper alignment of wound edges, avoidance of wound cavities, layered wound closure, and eversion of wound edges. Traumatization of the wound edges (bruising, laceration) by the surgeon should be avoided. Deep dermal sutures should be used to keep the skin margins as close together as possible without tension. To avoid wound dehiscence, abscesses at the puncture site, and suture granulomas, the subcutaneous sutures should not be too superficial. To prevent suture extrusion, asymmetric knot constructions should be avoided [12,20,21].

## 3. Biology of Wound Healing

Regardless of the origin of a wound, the healing process is composed of 4 phases: the exsudative, the resorptive, the proliferative, and the regenerative phase [5]. The different phases of wound healing influence each other and overlap [22].

After the injury of the skin, integument bleeding occurs. This activates the coagulation cascade. During the exsudative phase, platelets flow into the tissue and, together with fibrin, form a coagulum. Platelets secrete growth factors, which in turn activate macrophages and fibroblasts. The released growth factors (cytokines) cause an influx of cell structures and the resorptive phase alternates the exudative phase [4,5,22].

During the first and third day, monocytes and neutrophilic granulocytes migrate into the wound. Their task is to fight bacteria and break down destroyed tissue components. In addition, they secrete cytokines that activate macrophages, fibroblasts, and keratinocytes. Macrophages, in turn, stimulate fibroblast proliferation and angiogenesis [4,5,22,23].

Between the third and seventh day, fibroblast invasion with vascular proliferation occurs in the proliferative phase. A characteristic feature is the formation of granulation tissue [24]. From the wound edge, epidermal cells grow into the wound. The granulation tissue fills the tissue defect and forms a new extracellular matrix. Contraction of the wound accelerates wound closure. In parallel, re-epithelialization of the defect and scar formation takes place [4,5,22,23].

The fourth and final phase of wound healing is characterized by repair and remodeling of the healing wound. This phase is mainly characterized by collagen synthesis and collagen incorporation to thereby achieve remodeling of the scar tissue. In this last phase, particularly the final mechanical resilience of the scar is restored. Intact body surface consists of 80–90% collagen I and 10–20% collagen III. Early wound healing is dominated by collagen III, which gradually gains tensile strength as remodeling progresses. This leads to an increase in the content of collagen III. Increased production of collagen, fiber thickening and cross-linking increase the resilience of the scar [4,25].

## 4. Classification of Wounds

Wounds can be differentiated into clean wounds, clean–contaminated wounds, contaminated, and infected/dirty wounds, based on the degree of colonization or infection (Table 1). Uninfected operative wounds are defined as clean wounds where no contamination with bacteria from the respiratory, gastrointestinal, and genital tracts has occurred. In clean–contaminated operative wounds secretion from the respiratory, gastrointestinal, and genital tracts has entered under controlled conditions. Accidental, open, fresh wounds are classified as contaminated wounds. These include wounds with acute non-purulent inflammation or necrotic tissue without purulent secretion. Infected/dirty wounds are defined as old, traumatic wounds with avital and necrotic tissue, bacterial contamination, and clinical infection [26].

## 5. Wounds: Acute to Chronic

When examining a wound, the first step is an inspection. This includes photo documentation and sizing the wound by its length, width, and depth. Irrigation and disinfection of the wound and the wound’s margins is needed. The extent of trauma is estimated by detecting the involved anatomic structures [6]. Depending on the location, size, bioburden, and embedded anatomical structures, the complexity of a wound is defined. If an acute wound does not undergo a spontaneous healing process within 4 to 6 weeks, an acute wound is defined as chronic [27]. The complex pathophysiology of wound healing can be impeded by numerous disease processes. The results are chronic, non-healing “problem wounds”, causing the patient discomfort and distress. In addition, they culminate in exorbitant costs for health care systems due to prolonged wound treatments [28].

## 6. Avoiding Surgical Site Infections

From a surgical perspective, it is critical to keep the risk of surgical site infections (SSI) as low as possible. Generally, operative risk factors which contribute to a high risk for SSI are long operative times, increased blood loss and emergency cases [26]. Aseptic, sterile working conditions and techniques as well as the use of antimicrobial agents, hair removal, and pre-operative antibiotics minimize the risk for SSI [26]. Regarding the preparation of the surgical site, skin preparation solutions that include a combination of chlorhexidine gluconate (CHG) and alcohol are valued the most effective in eliminating skin contaminants. Alcohol is fast-acting whereas CHG provides a long residual effect in reducing microbes [29,30,31].

According to a review from Edmiston et al., hair removal is considered as preoperative and perioperative clipping without damaging the skin. In this way, bacterial contamination by existing hair and insufficient disinfection may be kept to a minimum [32].

The goal of preoperative antibiotic administration is to prevent wound infections and bacteremia caused by contaminating microorganisms during surgery. Systemic antibiotic prophylaxis should be administered within 60 min before incision. A first or second generation cephalosporine (i.e., Cefuroxim/Cefazoline) is recommended for routine surgical prophylaxis. They have a broad spectrum of activity, covering Gram-positive, aerobic Gram-negative and anaerobic Gram-positive germs. They have excellent distinction files in bone, muscle, and synovia. Preoperative antibiotics should always be administered to high-risk patients for endocarditis and to patients with pre-existing prostheses. They should not be continued for more than 24 h after surgery [33]. Systemic pre-operative antibiotic prophylaxis is recommended for clean-contaminated, contaminated, and dirty surgical sites of the head, neck, the upper limb, the hands, the skin, and for rhinoplasty. In addition, they are recommended for the reduction of SSI in clean breast plastic surgery [34].

## 7. Wound Treatment Principles: Debridement

A basic principle of wound treatment to achieve wound closure in acute and chronic wounds after exploration is initial adequate debridement [27]. Debridement is essential to good wound bed preparation [2,35]. The goal is to remove barriers such as necrotic tissue, senescent and non-migrating cells, bacteria, and foreign bodies that interfere with wound healing and impede the migration of keratinocytes across the wound bed [36]. Wound debridement can be conducted using five different techniques: mechanical, enzymatic, biological, autolytic, and surgical (Table 2) [2,23,37,38].

### 7.1. Autolytic Debridement

Autolytic debridement is carried out by the body itself with the aid of its own proteolytic enzymes and phagocytic cells. It is a slow and gentle process and may be enhanced by maintaining a moist wound environment which causes avital tissue to swell. Commonly used products are hydrogels. An example for a hydrogel is Prontosan^®^ wound gel. It contains polyhexanide which is a highly effective broad-spectrum antimicrobial against Gram-positive and Gram-negative bacteria. In autolytic debridement, vital tissue is liquefied by the body’s own enzymes (proteolysis) and flushed out of the wound with the exudate. Infected wounds are a contraindication to autolytic debridement [36,38,39].

### 7.2. Enzymatic Debridement

In enzymatic debridement, enzymatic agents break down necrotic tissue only in the wound bed. Viable tissue and the area surrounding the wound are left out [36,38]. An example is the enzymatic debridement of burn eschar with Nexobrid^®^. This enzyme mixture containing bromelain permits an early, selective removal of necrotic tissue in deep burn wounds by reducing the necessity for surgical treatment and allowing an early wound closure with wound dressings [40,41].

### 7.3. Mechanical Debridement

Mechanical debridement physically removes necrosis and debris from wounds. Mechanical debridement includes wet-to-dry dressings, wound irrigation with saline solution, and antimicrobial wound irrigation solution (Prontosan^®^ and Lavanox-Serag^®^ wound irrigation solution). Special wound irrigation solutions (Prontosan^®^, Lavanox^®^) prevent the growth of bacteria (such as *Pseudomonas aeruginosa*, MRSA) [39,42]. Mechanical debridement is most applicable to wounds with large amounts of devitalized tissue and does not distinguish between viable and nonviable tissue [36].

### 7.4. Biological Debridement

Biological debridement is an old, fast, and highly selective procedure that removes only necrotic tissue. Sterilized bottle fly maggots (Phaenicia sericata) are applied to wounds far from vital organs or underlying anatomical structures. According to several clinical trials, good results can be achieved especially in pressure ulcers, venous stasis ulcers, and neuropathic foot ulcers. The disadvantage of maggot therapy is patient discomfort [36].

### 7.5. Surgical Debridement

A surgical or sharp debridement is a very fast method. It is an appropriate technique to remove significant amounts of necrotic and infected tissue with sharp surgical instruments. At the pathophysiological level, bleeding occurs and thrombocytes occupy the wound space. Platelets activate coagulation and thus initiate the first phase of wound healing, the inflammatory phase. In chronic wounds, the natural physiological course of wound healing has come to a halt. The inflammatory phase is part of the usual wound healing process. Elevated levels of pro-inflammatory cytokines such as TNF-beta and IL-1 can be detected during a prolonged inflammatory phase in chronic, non-healing wounds. This is most likely due to the presence of bacteria and fungi in the open wound [43]. Sharp debridement and thus the elimination of bacterial colonization can transform chronic, protracted wounds back into acute wounds. Thus, the usual physiological course of wound healing is reintroduced [36,38]. In addition, sharp debridement makes it possible to obtain tissue for microbiological examination, especially in cases of suspected infection and bacterial colonization [36].

According to a retrospective cohort study by Wilcox et al., healing outcomes improve with the number and frequency of sharp debridements performed. In this study, more than 312,000 wounds were analyzed. Most of the wounds were diabetic foot ulcers, venous leg ulcers, and pressure ulcers [38]. In a clinical study by Saap and Falanga, adequate surgical debridement of diabetic foot ulcers showed better healing outcomes [44].

A special form of hydrosurgical debridement can be established through the use of the Versajet Hydrosurgery System (Smith and Nephew, Hull, UK). The Versajet Hydrosurgery system uses a moving stream of saline solution across a small window in an operating hand piece. The moving saline creates a vacuum cutting and aspirating tissue from the wound bed. The operator is able to regulate the tissue excision of the water jet by modifying its pressure and velocity. It is an easy and precise technique and facilitates maintaining the correct dermal plain [36,38,45].

## 8. Oxygen Supply and Bacterial Colonization

Surgical sharp debridement should be performed only in areas where local blood and oxygen supply are not compromised. An assessment of the local blood supply should be performed before performing surgical debridement, especially for lower extremity wounds [35]. If the regional pulse in the foot is not palpable, the blood flow is below 80 mmHg. Oxygen supply and oxygen tension in the wound bed are essential requirements for physiologic wound healing. Tissue oxygen tensions greater than 40 mmHg are required for proper wound healing. According to Sibbald et al., non-healing wounds are often hypoxic with low oxygen tensions (5–20 mmHg) due to poor blood supply [43]. Oxygen interacts with cytokines, cell proteins responsible for the proliferation of cells and cell migration. Therefore, when treating chronic wounds in patients with peripheral vascular disease and diabetic foot ulcers, a specific interdisciplinary approach is needed to address the underlying condition. Satisfactory wound healing cannot occur until the local blood supply is restored by revascularization [4,28].

Low oxygen supply in wounds leads to bacterial colonization, another point in the vicious cycle of wound healing disorders. Hypoxic conditions in wounds lead to cell death and tissue necrosis. In situations of arterial and venous insufficiency, trauma with blood loss, tissue injury, and edema, tissue perfusion is disturbed. These are ideal growing conditions for microorganisms arising from the surrounding skin, the oral and gastrointestinal tracts and the external environments. Particularly, anaerobes continue to proliferate simultaneously with other facultative bacteria, consuming the remaining oxygen [43].

Aerobic facultative bacteria such as Staphylococcus aureus, *Pseudomonas aeruginosa* and beta-hemolytic streptococci are common bacteria isolated in chronic wounds (Figure 2). Polymicrobial flora in wounds is widely blamed for delayed wound healing and infections [43].

## 9. Wound Closure Step by Step

The goal of treating wounds of acute and chronic–complex nature is to achieve unimpeded wound closure with a resilient scar. In both situations, wound healing can be achieved spontaneously, through primary and secondary wound healing, or with surgical reconstructive methods. Basic principles of wound closure, such as infection control, avoidance of mechanical stress, a normal nutritional status, diabetes control, and consistent phase-appropriate wound care, are essential components to achieve healing of acute and chronic wounds. An interdisciplinary approach is essential. In the case of a complex and chronic wound, it is necessary to establish professional wound management and care over the entire period [1,4].

### 9.1. The Reconstructive Ladder

To achieve a functional result, the reconstructive ladder (Figure 3) was developed as a guide for surgeons in wound management [14]. The principle of the reconstructive ladder was described by Mathes and Nahai in 1982. It describes a well-structured treatment algorithm with several wound closure techniques applicable to the different wound situations. Wound closure should be established with the simplest and, at the same time, the most effective method. According to the different steps of the reconstructive ladder, the surgical intervention effort increases with the wound’s complexity [8]. 

As a rule, the simplest method of wound treatment for acute and clean wounds without tissue loss is primary closure. More complex wounds can reach wound closure through tissue granulation. Granulation tissue arises from connective tissue which is surrounding a damaged area. At the cellular level it is composed mainly of small vessels, inflammatory cells, fibroblasts, and myofibroblasts. In case of wound closure, mainly myofibroblasts and vascular cells disappear. Through this mechanism of apoptosis, granulation tissue transforms into a scar [1,46,47].

In some cases, wounds will not granulate [48]. In cases with absent wound bed granulation, skin grafts, local, regional, and free flaps are further reconstruction options [49]. There are several options of grafts available for different wounds. Dependent on the wound’s localization and size, the plastic surgeon has to decide between full-thickness skin grafts (i.e., for small wounds in the face) or split-thickness skin grafts (STSG) [14]. Full-thickness skin grafts contain the epidermis and dermis. This prevents contraction after healing. They give better aesthetic results and are typically chosen for facial reconstruction. In skin graft healing, initial graft survival depends on osmotic transfer of oxygen and nutrients. These are delivered via the wound bed into the overlying graft. The thinner the graft, the easier the tendency of healing [50].

STSGs include the epidermis and varying layers of the dermis. They are suitable when resurfacing of large wounds is needed. They are frequently used to provide coverage in burn surgery. Furthermore, they are used for wound closure in soft tissue injuries and for diabetic foot and leg ulcers [50,51]. STSGs can be enlarged by multiple mechanical expansion. This type of skin graft, called a mesh graft, allows the coverage of a larger area per cm^2^ compared to the donor site. They also allow drainage of wound secretions through the numerous interstitial spaces [47]. STSG healing proceeds through three stages: anchorage, inosculation, and maturation. The first two stages are the most critical [52]. There are a few reasons why STSGs do not heal successfully in some cases. As with wound healing in general, there are several factors that interact and lead to inadequate healing. Factors that affect healing are the size of the wound, the anatomical location, the age of the patient, the blood supply to the wound bed, and the bacterial colonization [50]. Moreover, according to current studies, an increased BMI is strongly associated with STSG failure [53]. According to Penington et al., waist to hip ratio is associated with skin graft failure in the face and neck. Truncal obesity, hypertension, dyslipidemia, insulin resistance, and impaired glucose tolerance are symptoms of the widely spread popular disease “metabolic syndrome”. In animal models for metabolic syndrome, increased systemic level of inflammatory cytokines and prothrombotic factors lead to impaired wound healing [54]. Moreover, from a surgical perspective, it is critical to achieve a close adherence between the graft and the wound bed. Stabilization on the wound bed can be achieved by stapling the graft to the wound bed. In addition, compression of the graft on the wound bed is provided with fat gauze, compression bandages and Negative Pressure Wound Therapy (NPWT). When STSG are grafted to wounds near joints, temporary restriction of joint motion with splints is required to prevent shearing of the graft [50].

According to Younes and colleagues, there is an advantage in preparing the wound before transplantation with topical phenytoin. The use of topical phenytoin 10% ointment 2 to 8 weeks on the wound bed prior to grafting, in addition to standard wound bed preparation including debridement of the necrotic tissue, may lead to an enhancement of the graft survival [55]. Phenytoin (PHT) is usually used in epilepsy treatment. Gingival overgrowth is one of the side effects affecting a significant number of patients taking phenytoin. Further clinical studies are needed to determine whether there is a relationship between this stimulatory effect and the evaluation of phenytoin in wound healing [56]. According to studies, preparing the wound bed with platelet-rich plasma could represent a further improvement in wound healing after STSG transplantation. According to the clinical study by Schade and Roukis, the addition of PRP with its growth factors to the graft site may reduce the healing time to 90% or more. There were 13 patients included with well-controlled comorbidities in the study [52].

In the case of skin loss in large areas, particularly in major burns, micrografting is a convenient technique in order to reach wound closure. Micrografting was developed by C.P. Meek in 1958. Postage stamp split-thickness skin grafts are cut into small squares and placed on silk sheets which in turn are attached to the wound [57,58]. The labor-intensive Meek technique allows skin graft expansion up to tenfold [57].

### 9.2. Temporary Closure through NPWT (Negative Pressure Wound Therapy)

In most cases, NPWT is used to create tissue granulation followed by soft tissue reconstruction with various flaps [14,47,59]. NPWT is a method of achieving temporary wound closure in the event of significant tissue loss. With this method, an intermittent or continuous subatmospheric pressure is applied to the wound bed. The modality includes usually an open-pore surface dressing that enables the absorption of wound secretion, an adhesive foil to maintain the applied negative pressure, tubing, a collecting canister for the wound exsudate, and the vacuum unit. When negative pressure is applied, wound healing is stimulated by reducing interstitial edema and wound fluid, improving local circulation, stimulating proliferation of local cells, and reducing wound size by contracting wound edges. In most cases, NPWT is used to create tissue granulation, followed by soft tissue reconstruction with various flaps [36,60,61,62].

### 9.3. A Combination of Procedures Leading to Successful Wound Closure

Nevertheless, there are areas, or more often circumstances, where even flaps do not lead to sufficient wound healing. Instead of connecting and growing into the underlying tissue, they remain floating over the initial wound bed. In these cases, an old approach comes into question (Figure 4). After multiple NPWT treatments and failure of sole skin grafts as well as modern flaps, this approach to increase the chances of wound healing can be a combination of STSG with the technique of buried skin grafts (BSG). The method of BSG was renewed in 2004, where it was found to be promising for clinical application [63]. Further studies showed positive results concerning the treatment of perianal burn wounds and ulcers [64]. The production of BSGs can be performed through the Meek machine. A split thickness skin graft is harvested in a thickness of 0.2 mm and cut into small squares through the Meek machine. The skin pieces are placed with a pair of tweezers into the granulation tissue of the wound. Approximately 100 BSG are used per 5 × 5 cm^2^. A Meek skin graft is placed on top of the BSGs. Finally, an external compression dressing is placed for 10 days on top of the Meek plissés. Especially in gluteal, perianal, and sacral areas, sole skin grafts are jeopardized due to infections, hematomas, and seromas. This method allows a partial wound closure through the Meek grafts from the top and seems to stabilize the BSGs until the buried skin grafts migrate to the surface (Figure 5).

This combined procedure has been performed on a few patients in our Department of Plastic Surgery in recent years after all other wound care procedures had been unsuccessful. A future study to better understand the advantage of this technique would be desirable.

## 10. Conclusions

In order to reach a successful wound closure, a thorough understanding of the mechanism of wound healing is required. Plastic surgeons benefit from a deep understanding of wound physiology and anatomy. Applying the wound situation adapted principles for wound care and including an interdisciplinary approach, if necessary, may minimize complications and lead to better results. Further studies are needed to gain possible new insights into the promotion of wound healing. Especially in severely pre-diseased patients with wounds that are difficult to treat and those with limited wound healing, new findings and treatment methods would be of great benefit.

## Figures and Tables

**Figure 1 medsci-09-00045-f001:**
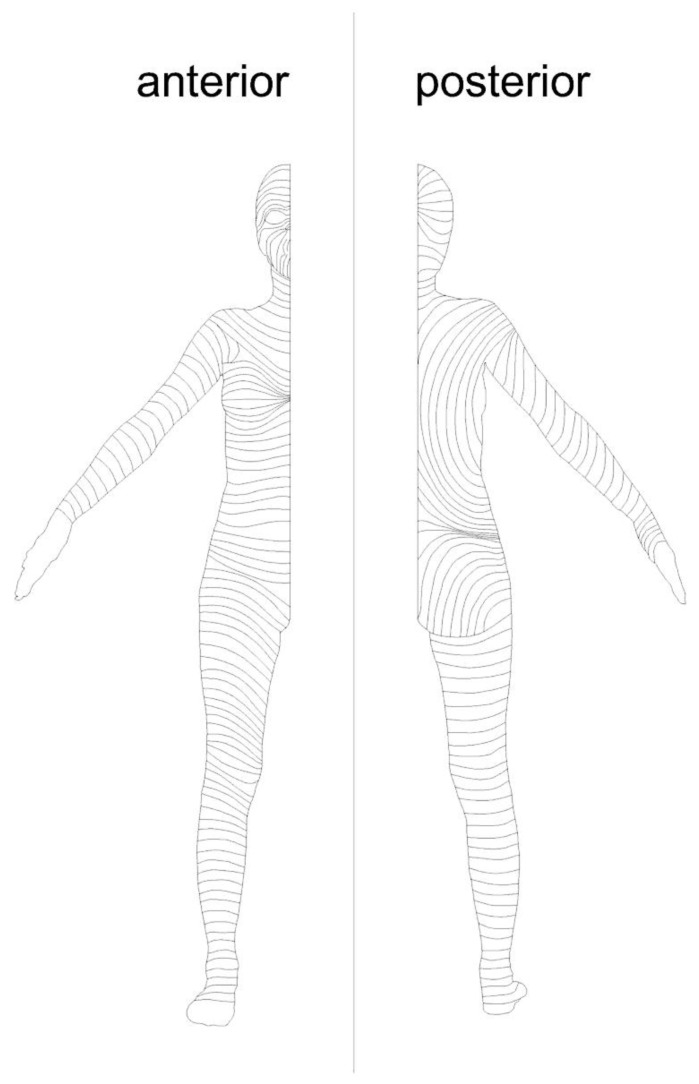
Relaxed skin tension lines.

**Figure 2 medsci-09-00045-f002:**
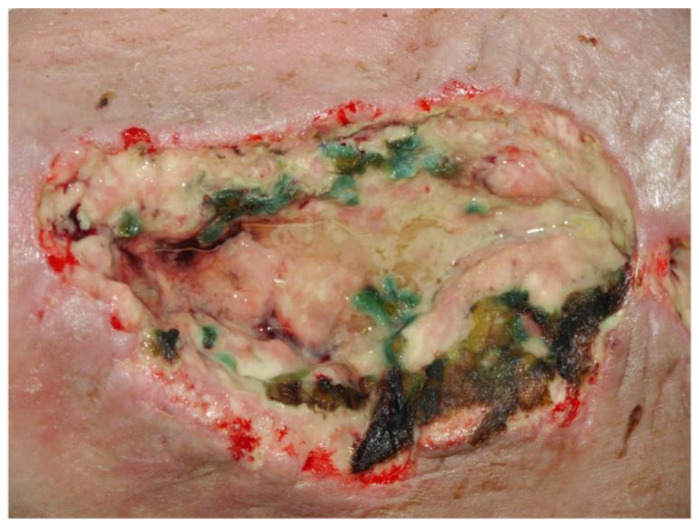
Wound colonized with *Pseudomonas aeruginosa*.

**Figure 3 medsci-09-00045-f003:**
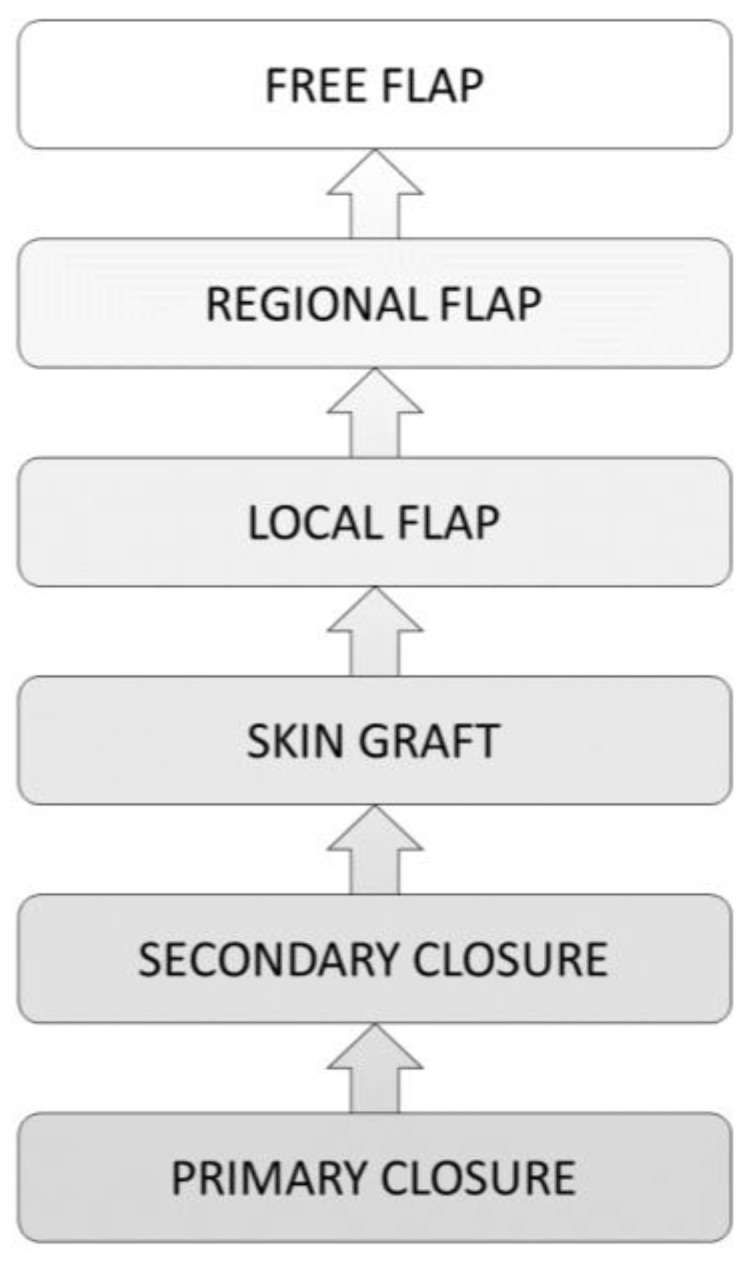
The reconstructive ladder.

**Figure 4 medsci-09-00045-f004:**
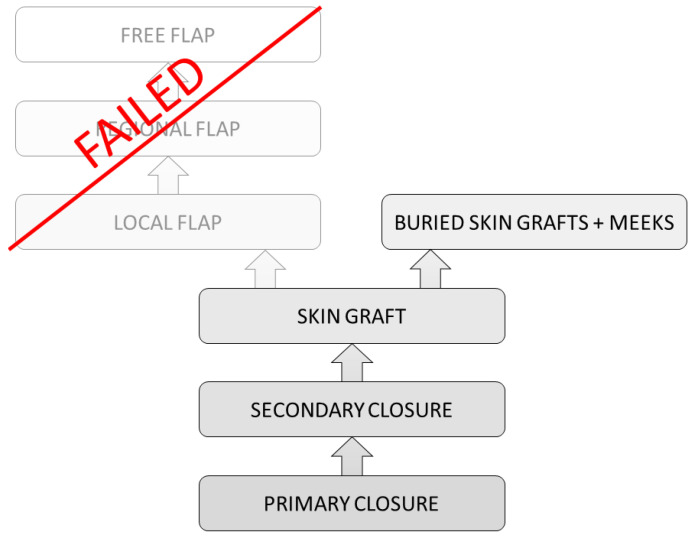
The reconstructive ladder: a combination of methods when conventional steps fail.

**Figure 5 medsci-09-00045-f005:**
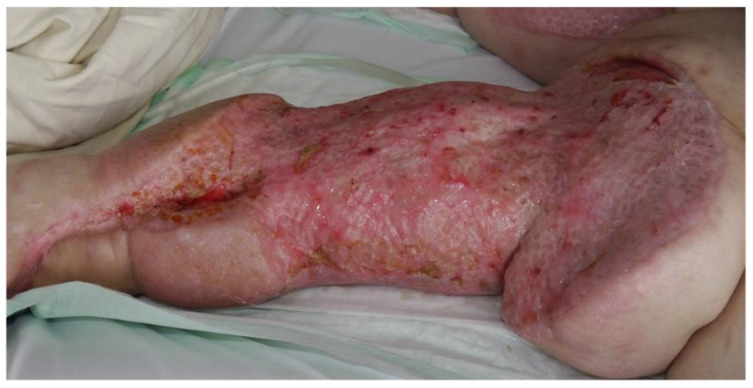
After wound closure by a combined method: Meeks and BSGs.

**Table 1 medsci-09-00045-t001:** Classification of wounds.

Wound Type	Characteristics
clean	uninfected operative incisional
clean/contaminated	uninfected secretion from the respiratory or gastrointestinal tract has entered
contaminated	accidental traumatic acute inflammation is possible no purulent secretion
infected/dirty	old necrotic, avital tissue purulent secretion

**Table 2 medsci-09-00045-t002:** Different types of debridement: advantages and disadvantages.

Debridement	Advantages	Disadvantages
autolytic	easy to perform	contraindicated for infected wounds
	natural	slow
	painless	
enzymatic	easy to perform	contraindicated for infected wounds
	highly selective	topical agents may inactivate enzymes
mechanical	easy to perform	non-selective
	faster than autolytic and enzymatic	can be painful
		may damage surrounding tissue
biological	highly selective	patient discomfort,
	fast painless	only for selected cases
surgical	fast,	applicable only with anesthesia,
	requires skilled clinician applicable for infected wounds	causes bleeding

## Data Availability

Not applicable.

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
