# Peer review of "Surgical Strategies to Promote Cutaneous Healing"

_medsci, 2021, doi:10.3390/medsci9020045_

Round 1

Reviewer 1 Report

The aim of the authors is commendable. However, some crucial weak points are present:

1)In the text, the concept of surgical wound and surgical care of  wounds due to other causes (e.g. Diabetic ulcers) are often merged creating confusion to the readers;

2) Which methodology used the authors used to do the literature search?Please add a short paragraph about it

3) High quality sources, such as Who Guidelines or Cochrane review are not cited, many statements are not referenced and there is no evaluation of the quality of evidence supporting the core concepts the authors highlighted in the text;

4) The absence of Immage/figures or tables about important topic (e.g. wound classification,, Langer's line) is another weak point

Reviewer 2 Report

The manuscript offers an extensive narrative review on skin healing in association with surgical strategies. What is missing from the text is a paragraph on the biology of skin healing. This is crucial for understanding this delicate process. It can also be useful in additional surgical strategies.
